# Heterologous Systemic Prime–Intranasal Boosting Using a Spore SARS-CoV-2 Vaccine Confers Mucosal Immunity and Cross-Reactive Antibodies in Mice as well as Protection in Hamsters

**DOI:** 10.3390/vaccines10111900

**Published:** 2022-11-10

**Authors:** Paidamoyo M. Katsande, Leira Fernández-Bastit, William T. Ferreira, Júlia Vergara-Alert, Mateusz Hess, Katie Lloyd-Jones, Huynh A. Hong, Joaquim Segales, Simon M. Cutting

**Affiliations:** 1Department of Biological Sciences, Royal Holloway University of London, Egham TW20 0EX, UK; 2Unitat Mixta d’investigació IRTA-UAB en Sanitat Animal, Centre de Recerca en Sanitat Animal (CReSA), Campus de la Universitat Autònoma de Barcelona (UAB), 08193 Bellaterra, Spain; 3Institut de Recerca i Tecnologia Agroalimentàries (IRTA), Programa de Sanitat Animal, Centre de Recerca en Sanitat Animal (CReSA), Campus de la Universitat Autònoma de Barcelona (UAB), 08193 Bellaterra, Spain; 4SporeGen Ltd., London Bioscience Innovation Centre, 2 Royal College Street, London NW1 0NH, UK; 5Departament de Sanitat i Anatomia Animals, Facultat de Veterinària, UAB, 08193 Bellaterra, Spain

**Keywords:** COVID-19, SARS-CoV-2, nasal vaccine, prime boost

## Abstract

*Background*: Current severe acute respiratory syndrome coronavirus 2 (SARS-CoV-2) vaccines are administered systemically and typically result in poor immunogenicity at the mucosa. As a result, vaccination is unable to reduce viral shedding and transmission, ultimately failing to prevent infection. One possible solution is that of boosting a systemic vaccine via the nasal route resulting in mucosal immunity. Here, we have evaluated the potential of bacterial spores as an intranasal boost. *Method*: Spores engineered to express SARS-CoV-2 antigens were administered as an intranasal boost following a prime with either recombinant Spike protein or the Oxford AZD1222 vaccine. *Results*: In mice, intranasal boosting following a prime of either Spike or vaccine produced antigen-specific sIgA at the mucosa together with the increased production of Th1 and Th2 cytokines. In a hamster model of infection, the clinical and virological outcomes resulting from a SARS-CoV-2 challenge were ameliorated. Wuhan-specific sIgA were shown to cross-react with Omicron antigens, suggesting that this strategy might offer protection against SARS-CoV-2 variants of concern. *Conclusions*: Despite being a genetically modified organism, the spore vaccine platform is attractive since it offers biological containment, the rapid and cost-efficient production of vaccines together with heat stability. As such, employed in a heterologous systemic prime–mucosal boost regimen, spore vaccines might have utility for current and future emerging diseases.

## 1. Introduction

The coronavirus infectious disease 2019 (COVID-19) pandemic will be remembered both for its impact on global health and society but also for the remarkable speed at which prophylactic vaccines were implemented. Vaccines (mRNA, viral-vectored and subunit) have all shown high levels of protection against the most severe forms of disease, reducing morbidity and mortality [1]. On the other hand, it has become apparent that current COVID-19 vaccines are unable to prevent person-to-person transmission of its causal agent, severe acute respiratory syndrome coronavirus 2 (SARS-CoV-2), with breakthrough infections observed in fully vaccinated individuals supporting this [2,3]. The dissemination of the virus will undoubtedly continue as a) immunity wanes and b) the reality that a significant portion of the world’s population (~40–45%) has yet to receive their first vaccine dose [1]. The longer the virus continues to circulate, the greater the probability that new variants of concern (VOCs) will emerge [1]. The poor durability of current vaccines with protection of about six months may necessitate repeated booster campaigns. Taken together, the COVID-19 pandemic cannot yet be considered closed and will continue to represent a major burden to public health authorities until better vaccines are developed.

SARS-CoV-2 transmits person-to-person by respiratory droplets and infects the upper respiratory tract (URT). Mucosal immunity is required to neutralize SARS-CoV-2 at the mucosal surfaces of the URT, either by tissue-resident memory T cells (TRMs) or a secretory IgA (sIgA) that can neutralize or prevent virus attachment within the epithelial cells of the mucosa. [4,5]. Vaccines administered by a parenteral route (noting that all COVID-19 vaccines so far are delivered by intra-muscular injection) fail to evoke a significant mucosal immune response, probably accounting for their failure to prevent the transmission and dissemination of the virus [6].

Accordingly, there is now a focus on developing mucosal COVID-19 vaccines of which intranasal is the optimal route of administration [7,8]. Mucosal vaccination has the potential to provide robust protective immunity at the mucosal site of pathogen entry by recruiting antigen-presenting cells (APCs) and/or engaging other innate immune cells. This requires safe and effective mucosal vaccination strategies that utilize both mucosal adjuvants and appropriate delivery systems [7]. SARS-CoV-2 vaccine peptides and a combination of flagellin (KF) and cyclic GMP-AMP (cgGAMP) as an adjuvant delivered intranasally have been shown to elicit strong systemic and mucosal immune responses that provide protection against SARS-CoV-2 challenge in a transgenic mouse model [9]. Early clinical data have suggested that without pre-existing immunity to SARS-CoV-2, immunity resulting from intranasal vaccination is suboptimal [7,10]. One promising approach might be that of boosting a current COVID-19 vaccine with an intranasal vaccine [11]. Adenoviral 5 and 19a vectored vaccines administered intranasally after a systemic mRNA vaccine prime were shown to induce high levels of both mucosal IgA and T_RM_, be fully protective in a mouse model of SARS-CoV-2 infection and neutralize VOCs [11].

Bacterial endospores of *Bacillus subtilis* engineered to display heterologous antigens have been shown to provide several advantages for mucosal immunization with efficacy against a number of bacterial and viral pathogens [12,13,14]. Spores carry natural mucosal adjuvant properties [15] and induce innate immunity [16]. Using a recently developed system (THY-X-CISE^®^), engineered spores can be rendered unable to proliferate in the environment ensuring, biological containment [17]. Spores are of uniform size (1 μm diameter), can be produced rapidly in large quantities and are currently manufactured at medicinal grade by several pharmaceutical companies [18]. Most importantly, spores are heat stable (<70 °C), and with a few exceptions, antigens expressed on their surface are stable both to heat and gastric enzymes [19].

Here, we engineered *B. subtilis* spores to express SARS-CoV-2 antigens and used these to nasally boost a systemic prime consisting of either a recombinant Spike protein or the Oxford AZD1222 vaccine. Our work shows that this heterologous systemic prime–mucosal spore boost strategy evokes mucosal immunity and is potentially protective.

## 2. Materials and Methods

### 2.1. General Methods

Methods for *B. subtilis*, including the preparation of spores and the extraction of spore coat proteins, are described elsewhere [20]. *B. subtilis* strain PY79 is a prototrophic laboratory strain.

### 2.2. Construction of B. subtilis Spores Expressing SARS-CoV-2 Antigens

A cloning method referred to as THY-X-CISE^®^ [17] was used to introduce heterologous genes into the chromosome of *B. subtilis*. First, chimeric genes were synthesized (Azenta Life Sciences), carrying the 5′ segment (including promoter) of either the *cotB* or *cotC* genes of *B. subtilis* fused at their 3′-end to DNA-encoding SARS-CoV-2 domains. Gene sequences are shown in Appendix A, which included the receptor-binding domain (RBD) (Wuhan-Hu-1 2019-nCov and Omicron B.1.1.529 variants) and the HR1-HR2 domain (Wuhan-Hu-1 2019-nCov) of the Spike protein. For clarity, these are referred to henceforth as S^Wuh^, S^Omi^ and HR1-HR2^Wuh^. Chimeras were then subcloned into the plasmid pThyA [17]. This placed the chimeric gene between the left and right arms of the *thyA* gene that encodes thymidine synthetase A (Appendix A). The linearization of the resulting plasmid and introduction into the *B. subtilis* chromosome (strain PY79) by double-crossover recombination generated trimethoprim-resistant transformants that were thymine-dependent. The resulting *thyA::insertion* strain was then used as a recipient in a second transformation where an empty pThyB vector was introduced, thus disrupting the *thyB* locus (*thyB::Δ*). The resulting strain (*thyA::insertion thyB::Δ*) was thymine-dependent but resistant to a higher concentration of trimethoprim. The strains constructed were PK120 *thyA::cotB-RBD^Wuh^ thyB::Δ*, PK122 *thyA::cotC-HR1-HR2^Wuh^ thyB::Δ* and PK230 *thyA::cotB-RBD^Omi^ thyB::Δ*. An isogenic strain, PK118 (*thyA::Δ thyB::Δ*), which carries no gene insertions was made using the same procedure with empty pThyA and pThyB vectors that carry no gene insertions. PK118 (*thyA::Δ thyB::Δ*), referred to henceforth as WT, was used as an isogenic control strain that produces “naked” spores devoid of SARS-CoV-2 antigens. Strains were confirmed by preparing spores, extracting spore coat proteins and probing Western blots with antibodies to the relevant SARS-CoV-2 domains: anti-S^Wuh^ PAbs raised against the entire S^Wuh^ polypeptide (SinoBiological, 40589-T62) and anti-S^Omi^ MAbs that were raised against the RBD of the S^Omi^ (SinoBiological, 40592-MM117).

### 2.3. Immunogens

Recombinant Spike protein rS^Wuh^ and rS^Omi^ (baculovirus expressed) were obtained from SinoBiological (Cat Nos. 40589-V08B1 and 40589-V08B33, respectively). Prior to intramuscular injection, 5 μg of protein was suspended in PBS and complexed (1:1) with the adjuvant AddaVax (Invivogen, vac-adx-10). The COVID-19 vaccine ChAdOx1 nCoV-19 (AZD1222) with a concentration of 3.8 × 10^9^ infectious units (IU)/mL (3.7 × 10^11^ virus particles/mL) was provided by Prof. Sarah Gilbert (Jenner Institute, Oxford University) and referred to henceforth as AZD1222. Prior to use, AZD1222 was diluted in PBS to give a working concentration of 2 × 10^9^ IU/mL.

### 2.4. Prime Boost Vaccination Studies

BALB/c female mice were obtained from Charles River, UK (aged 8–10 weeks before study start) and allocated into groups (n = 5–6) for recombinant prime and n = 6 for the AZD1222 prime.

(a) Recombinant prime boost vaccination (Appendix A)

Group 1 received no immunizations and served as a naïve control group. Groups 2–4 each received an intramuscular injection (i.m.) of 50 μL (5 μg) of formulated rS^Wuh^ protein in each hind quadricep muscle. Group 2 was primed (i.m.) with rS^Wuh^ on day 1 and culled on day 48. Groups 3 and 4 were primed (i.m.) with rS^Wuh^ on day 1 and then intranasally (i.n.) boosted three and five weeks later with spores (1 × 10^9^ CFU) of either WT (‘naked’) spores (Group 3) or a 1:1 mixture of PK120 (CotB-RBD^Wuh^) and PK122 (CotC-HR1-HR2^Wuh^) referred to as SporCoVax (Group 4). Each intranasal administration (10 μL total/animal, 5 μL/nare, no anesthesia) consisted of three administrations given on consecutive days (Boost 1, days 21, 22 and 23 and Boost 2, days 35, 36 and 37).

(b) AZD1222 prime boost vaccination (Appendix A)

Group 1 was a naïve group and received no immunizations. Groups 2–5 each received 50 μL of AZD1222 (1.0 × 10^8^ IU) in phosphate-buffered saline (PBS) in the right hind quadricep muscle. Group 2 was culled on day 48. Groups 3–5 were boosted (intranasal; 5 μL/nare, no anesthesia) three and five weeks later with spores (1 × 10^9^ CFU) of WT (‘naked’; Group 3), SporCoVax (a 1:1 mixture of CotB-RBD^Wuh^ and CotC-HR1-HR2^Wuh^ spores; Group 4) or PK230 (CotB-RBD^Omi^; Group 5). Each intranasal administration (10 μL total/animal, 5 μL/nare, no anesthesia) consisted of three administrations given on consecutive days (Boost 1, days 21, 22 and 23 and Boost 2, days 35, 36 and 37).

### 2.5. Determination of Mucosal and Systemic Antibody Titers by Indirect ELISA

For the analysis of immunological responses, saliva was collected on days −1, 20, 34 and 48, serum was taken on day 48, and lungs were collected on day 48 and stored at −80 °C. For lungs, sample extractions were made at a one-fifth (*w*/*v*) dilution in extraction buffer (2% (*v*/*v*) fetal calf serum) containing protease inhibitors, EDTA (0.05 mg/mL), as previously described [21]. Saliva extractions were made at a one-tenth (*w*/*v*) dilution in PBS. Samples were gently shaken for 2 h at 4 °C to disrupt solid material, centrifuged (8000× *g*, 15 min.) and the supernatant used for analysis. Antibody levels in saliva (sIgA), lungs (IgA) and serum (IgG) were quantified by indirect enzyme-linked immunosorbent assay (ELISA). Greiner 96-well plates (MaxiSorp) were coated with 2 µg/mL of either rS^Wuh^ or rS^Omi^ (50 μL/well) in PBS overnight at 4 °C, followed by blocking for 1 h at RT with PBS containing 2% (*w*/*v*) bovine serum albumin (BSA). Saliva samples were diluted 1:20 in PBS. Lung and serum samples were diluted 1:10 and 1:1000, respectively, in diluent buffer (0.01 M PBS, 1% (*w*/*v*) BSA, 2% (*v*/*v*) FBS, 0.1% (*v*/*v*) Triton X-100, 0.05% (*v*/*v*) Tween 20). Samples were added to plates and 2-fold serially diluted. Plates containing saliva and lung samples were incubated for 2 h at 30 °C, and those containing serum samples were incubated for 2 h at RT. Levels of IgA and IgG were detected using the appropriate horseradish peroxidase-conjugated anti-mouse IgA (Sigma Cat No. A4789-1) or anti-mouse IgG (Dako Cat No. P0447) in conjugate buffer (2% (*v*/*v*) FBS, 1% (*v*/*v*) BSA, 0.05% (*v*/*v*) Tween 20 in 0.01 PBS). Plates were incubated for 1 h at RT and then developed using tetramethyl benzidine (TMB) substrate (0.1 mg/mL 3.3′,5.5′-tetramethylbenzidine in 0.1 M sodium acetate buffer (pH 5.5)). Reactions were stopped using 2 M H_2_SO_4_, and the ODs were read at 450 nm. Dilution curves were created for each sample, and endpoint titers were estimated as the maximum dilution that gave an absorbance reading above the average naïve sample.

### 2.6. Cytokine Analysis

Mouse spleens were harvested, homogenized and treated with ACK lysis buffer (Sigma Cat. No 11814389001) to remove erythrocyte contamination. Splenocytes were cultured at 1 × 10^6^/well in 96-well plates. Cells were stimulated with 2.5 µg/mL of rS^Wuh^ for 72 h. IL-2, IL-5 and TNF-α in culture supernatants were measured using an 8-plex multiplex immunoassay (Biolegend Cat No. 741053) according to the manufacturer’s instructions. Data were acquired using a BD FACSCanto™ II, and analysis was performed using Legendplex software (Biolegend, San Diego, CA, USA).

### 2.7. Hamster Challenge Study

Golden Syrian hamsters (age 6–8 weeks, 80–120 g) were allocated into groups (50% male, 50% females). Group 1 (n = 6) was a naïve, ‘unchallenged’ control group. Animals were dosed with PBS buffer on day 1 (i.m.), day 22 (i.n.) and day 35 (i.n.) and culled on day 49. Group 2 (n = 12) was a negative control group dosed as Group 1 but challenged with SARS-CoV-2 on day 42 and culled on day 49. Group 3 (n = 12) was primed (i.m.) on day 1 with rS^Wuh^ (10 μg, formulated 1:1 with the AddaVax adjuvant) and then boosted (intranasal) on days 22 and 35 with SporCoVax (a 1:1 mixture of CotB-RBD^Wuh^ and CotC-HR1-HR2^Wuh^ spores at 2.5 × 10^9^ CFU), after which, animals were challenged with SARS-CoV-2 on day 42 and culled on day 49. For intranasal administration, a total volume of 0.1 mL/dose (50 μL/nare) was used, and procedures were conducted under anesthesia (isofluorane).

A SARS-CoV-2 Cat02 isolate (variant D614G, ID EPI_ISL-47147) was used for challenge. It was isolated from a patient in a laboratory-confirmed COVID-19 case in Barcelona, Spain, and propagated in Vero E6 cells (ATCC CRL-1586). The inoculum was prepared on the same day of the challenge at a concentration of 10^5^ TCID_50_/mL and kept refrigerated until use. Then, 0.1 mL was administered per challenge dose (50 μL/nare under isofluorane anesthesia), giving a challenge dose of 10^4^ TCID_50_. At 2 and 4 days post challenge, 4 animals/group were sacrificed for necropsy, and at day 7, all remaining animals were culled. Methods for qPCR in nasal turbinates (NT), oropharyngeal (OP) swabs, etc., were performed as described elsewhere [22].

### 2.8. Statistical Analysis

Statistical significance was assessed by Mann–Whitney U test. All statistical analyses were performed using Prism (GraphPad, San Diego, CA, USA). Flow cytometry data were analyzed using LEGENDplex™ Data Analysis Software.

## 3. Results

### 3.1. Construction of B. subtilis Spores Displaying SARS-CoV-2 Domains

Isogenic strains of *B. subtilis* were engineered to express SARS-CoV-2 (Wuhan-Hu-1 variant 2019-nCov) antigens on the spore surface as chimeric fusions to the spore coat proteins CotB or CotC. This included two segments of the Spike (S) protein, the RBD and the HR1-HR2 domains fused to CotB (PK120, CotB-RBD^Wuh^) and CotC (PK122; CotC-HR1-HR2^Wuh^). A third strain, PK230, was made which expressed the RBD of the Omicron variant (B.1.1.529) fused to CotB (CotB-RBD^Omi^).

*B. subtilis* vaccine strains carrying insertions at the *thyA* loci were examined by Western blotting of SDS-PAGE size-fractionated spore coat proteins extracted from preparations of pure spores (approx. 2 × 10^9^ spores/extraction). Blots were probed with PAbs or MAbs as shown. 

The use of anti-S^Wuh^-specific PAbs Western blotting of proteins extracted from purified spores of each strain showed that the RBD (primary CotB-RBD^Wuh^ species of ~62 kDa) and HR1-HR2 (CotC-HR1-HR2^Wuh^, 40 kDa) domains were expressed on the spore surface. Proteins extracted from spores of an isogenic WT strain devoid of SARS-CoV-2 antigens exhibited no cross-reaction (Figure 1A). Using an Omicron-specific anti-S^Omi^ MAb expression of RBD was apparent in spores of PK230 as a primary CotB-RBD^Omi^ species of ~63 kDa (Figure 1B).

### 3.2. Intranasal Boosting with a Spore Vaccine following a Systemic Prime Evokes Mucosal Immune Responses

We used a heterologous systemic prime–mucosal boost regimen to determine whether the intranasal administration of spore vaccines expressing SARS-CoV-2 antigens could evoke mucosal immunity. Mice were first dosed by intra-muscular injection (i.m.) with either the rS^Wuh^ protein or the AZD1222 vaccine and then, 21 and 35 days later, administered spore vaccines as an intranasal boost. For murine studies, a single dose consisted of three daily intranasal administrations (e.g., days 21, 22 and 23) and resulted from the regulatory dosing limitations required for mice. Following a prime of rS^Wuh^ or AZD1222 animals administered SporCoVax, a 1:1 mixture of CotB-RBD^Wuh^ and CotC-HR1-HR2^Wuh^, spores exhibited antigen-specific sIgA in the saliva together with seroconversion (Figure 2A,B). sIgA levels were significantly (*p* < 0.001) greater than the group receiving only an rS^Wuh^ or AZD1222 prime, each of which yielded essentially no detectable sIgA (Figure 2A,B). Interestingly, isogenic spores (WT) carrying no SARS-CoV-2 antigens also evoked antigen-specific sIgA but at levels significantly lower (*p* < 0.01) than in animals dosed with SporCoVax (Figure 2A,B). Antigen-specific IgA was also detectable in lung extracts (day 48) of animals primed only with rS^Wuh^ or AZD1222, as well as those primed and then boosted with WT spores, but levels were significantly higher (*p* < 0.01) in mice boosted with SporCoVax (Figure 2C,D). The presence of IgA in lung extracts of mice dosed only with rS^Wuh^ or AZD1222 (i.m.) most probably results from contamination with serum IgA and should be considered here as a baseline. Serum IgG levels measured (Figure 2E,F) at day 48 showed significantly (*p* < 0.01) higher levels of rS^Wuh^-specific IgG in animals primed and then boosted with SporCoVax spores compared to animals primed only or primed and then boosted with WT spores (*p* < 0.01). 

Using a similar systemic prime–mucosal boost regimen, we primed (i.m.) mice with the AZD1222 vaccine and boosted with either WT spores (no antigens) or PK230, a spore vaccine expressing CotB-RBD^Omi^. As reported above for boosting with spores displaying Wuhan-specific antigens, boosting with CotB-RBD^Omi^ spores evoked antigen (rS^Omi^)-specific sIgA in saliva and serum which was at significantly higher levels than in groups dosed with WT spores (*p* < 0.001 and *p* < 0.01, respectively) or in animals primed only (*p* < 0.01) (Appendix A).

Together, these results show that intranasal boosting with spores expressing SARS-CoV-2 antigens following a systemic prime evokes mucosal immunity.

### 3.3. Boosting with Spores Evokes Cross-Reactive Antibodies

From the study above in which animals had been primed with AZD1222 and then boosted with SporCoVax, we determined whether saliva samples carried antibodies that cross-reacted with the Omicron variant. Using ELISA plates coated with rS^Omi^, we showed that SporCoVax-boosted animals exhibited antigen-specific sIgA in saliva samples which was significantly greater (*p* < 0.01 and *p* < 0.001, respectively) than in animals dosed with WT spores or rS^Wuh^ only (Figure 3). As shown in the previous section, WT spores used alone as a boost also enhanced antigen-specific sIgA.

### 3.4. Intranasal Boosting with a Spore Vaccine Results in a Mixed T Cell Cytokine Profile

We next studied which subsets of T helper (Th) cells and cytotoxic T cells were responding to the rS^Wuh^ protein. Animals (n = 3/group) were either primed with AZD1222 or primed (AZD1222) and then boosted with WT or SporCoVax spores. Splenocytes from immunized animals were cultured with rS^Wuh^ and assessed for the production of IL-2, TNF-α and IL-5, which are induced in Th1 (IL-2, TNF-α) and Th2 (IL-5) T-cell responses (Figure 4). SporCoVax-boosted animals exhibited marked increases in the expression of all three cytokines compared to other groups. Together, this shows that intranasal spore boosting confers both Th1 and Th2 responses.

### 3.5. Intranasal Boosting with Spores Is Protective in the Hamster SARS-CoV-2 Model of Infection

The golden Syrian hamster model is currently considered one of the best models to study the effects of SARS-CoV-2 infection because it permits high levels of SARS-CoV-2 replication and generates significant clinical signs (weight loss) and moderate to severe pneumonia [23,24]. To determine whether intranasal boosting with spores expressing SARS-CoV-2 antigens conferred protection, we first primed animals (n = 12; Group 3) with rS^Wuh^ protein (i.m.) and then administered two intranasal boosts of SporCoVax. Seven days after the last boost, animals were challenged (intranasal) with SARS-CoV-2 (10^4^ TCID_50_/animal). Two additional groups, Group 1, being a naïve, unchallenged group (n = 6), and Group 2, a negative control group (n = 12) which received PBS in place of immunogens (i.m. prime and two i.n. boosts) and challenged with SARS-CoV-2 seven days after the last dose, were used for comparison. 

Golden Syrian hamsters were primed (i.m.) with recombinant Spike (rS^Wuh^) protein followed by two intranasal boosts with 2.5 × 10^9^ spores of SporCoVax (1:1 mixture of CotB-RBD^Wuh^ and CotC-HR1-HR2^Wuh^ spores) (Group 3). Negative, unvaccinated (Group 2) and naïve (Group 1) control groups received PBS in place of immunogens. Seven days after the final boost, animals in Groups 2 and 3 were intranasally challenged with SARS-CoV-2 (D614G variant, 10^4^ TCID_50_/animal). Weight loss post challenge was shown (panel A). Animals were sequentially euthanized (2, 4 and 7 dpi), necropsies were performed, and viral loads of SARS-CoV-2 were determined by qPCR in nasal turbinate (panel B), OP swabs (panel C) and lungs (panel D). Mann–Whitney, * *p* < 0.05, ** *p* < 0.01.

Weight loss was used to track the progression of the disease (Figure 5A). Naïve (Group 1) animals showed no reduction in weight, whereas non-vaccinated animals (Group 2) challenged with SARS-CoV-2 showed progressive reductions in weight 2–7 dpi. By contrast, animals from Group 3 (SporCoVax boost) lost weight at 2 dpi and started to recover at 4 dpi, and finally regained 100% of their initial weight at 7 dpi (*M* = −7.4, *SE* = 1.73, 95%. *CI* [−11.65, −3.2], *p* < 0.01). For operational reasons, we were unable to assign animals to a prime-only group. We acknowledge this as a major flaw in the experimental design and interpretation. On the other hand, we note that it is extensively reported that a single systemic dose of S protein fails to induce protective antibodies [25,26,27].

To assess the impact of intranasal boosting on viral replication, we examined SARS-CoV-2 RNA (gRNA) in NT (Figure 5B), OP swabs (Figure 5C) and lungs (Figure 5D) at 2, 4 and 7 dpi. After SARS-CoV-2 inoculation (Groups 2 and 3), viral gRNA was detected at 2 dpi in NT, OP swabs and lungs in both groups. The viral load decreased progressively in OP swabs and both tissues in animals receiving intranasal boosts with SporCoVax (PK220 boost; Group 3), compared to the unimmunised–infected control group (Group 2), which was statistically significant at 7 dpi (*p* < 0.01). Taken together, we conclude that the heterologous systemic prime–intranasal boost regimen ameliorates the clinical and virological outcome of SARS-CoV-2 challenge.

## 4. Discussion

The continued dissemination of SARS-CoV-2 ensures that COVID-19 is likely to persist. To halt transmission, vaccine strategies must focus on blocking interaction of the virus at the mucosa. Accomplishing this requires mucosal immunity and more specifically induction of sIgA that can block viral adhesion. We present evidence that bacterial spores engineered to express SARS-CoV-2 antigens and delivered as an intranasal boost following a systemic prime (either a S protein subunit or the AZD1222 COVID-19 vaccine) induce antigen-specific sIgA. This is further supported by a robust IL-5 response often associated with sIgA induction [5]. Moreover, when this strategy was employed in hamsters, they showed signs of protection from viral challenge including recovery of body weight and a reduction in viral shedding. This is encouraging for a number of reasons but most importantly a reduction in viral shedding would lead to reduced transmission and dissemination ultimately providing a means to controlling this disease. A heterologous systemic prime – mucosal boost vaccination strategy is being considered as one of the most promising routes forward to enhance existing COVID-19 vaccines [11,28]. The underlying basis for how a mucosal boost induces a localized response is not yet fully understood but a “prime-pull” mechanism has been proposed where systemic memory cells (induced by the prime) are expanded during the recall response and which then migrate to the mucosa followed by differentiation into tissue-resident memory cells [29]. 

Spores are dormant entities that are found in soil and air and as such in humans and animals also [30]. As dominant members of the aerobiome (~10%) humans are continuously exposed to *Bacillus* spores with approximately 10–10^5^ spores inhaled daily [31]. In the gastro-intestinal tract they are present at constant levels of ~10^4^ spores/g of feces [32]. The spore vaccine platform has been well documented, and spores genetically engineered or, alternatively, adsorbed with heterologous antigens, evoke balanced Th1/Th2 responses as well as augmenting mucosal antibody responses [15]. This is supported by the studies here using a genetically engineered spore vaccine and resulting antibody and cytokine responses. Using the THY-X-CISE^®^ cloning, spores, should they germinate, are unable to proliferate in the environment since live cells die immediately due to a ‘thymine-less’ death [17]. Biological containment is thus ensured, and a similar strategy has been used to enable clinical evaluation of a *Lactococcus lactis* IL-10 delivery system [33]. Spore vaccines can be constructed rapidly on a case-by-case basis although, as shown here, antibodies resulting from a Wuhan-delivered antigen cross react with the Omicron variant suggesting that the vaccines described here might be protective against emerging VOCs. Spores are themselves heat stable and can be stored indefinitely in an aqueous or desiccated form. Of course, this may not necessarily apply to a heterologous protein displayed on the spore surface. Other spore vaccines we have generated do indeed show stability, but this would need to be addressed by an extensive program of formulation and stability studies.

Intriguingly, nasal boosting with isogenic spores displaying no antigens also showed a clear augmentation of mucosal antigen-specific responses. The underlying mechanism is not fully understood but other studies have shown that *B. subtilis* spores carry natural adjuvant properties [15,34]. Spores can bind antigens including intact virions and adjuvanticity is observed when spores and adjuvants are administered together or by separate routes (systemic-mucosal) [15,16,19]. Spores have also been shown to act as immunomodulators and specifically by inducing innate immunity [16]. Spores can be taken up by M-cells [35], persist within phagocytes [36,37] and intriguingly, inactivated spores are also able to stimulate the innate immune system when delivered nasally [16]. This is achieved by interaction of spores with Toll-like receptors (TLR2 and TLR4), cytokine induction, recruitment of NK cells and maturation of Dendritic cells [16,37]. What is interesting is that here we show that ’naked’ spores administered three weeks after the systemic prime are able to augment mucosal immunity suggesting that spore-induced innate immunity may be a factor. Although the mucosal responses are clearly greater when spores-displaying antigens are delivered the use of non-recombinant spores may, long-term, have greater utility. It is worthwhile noting that in previous work we have shown that nasal administration of spores alone is sufficient to confer protection against influenza (H5N2) using both murine and ferret models of infection [16,38]. Based on the observed seroconversion of sIgA in mice we suspect that a single dose of spore vaccine using this regimen (3-weeks post-prime) is unlikely to be protective but further empirical analysis of dosing regimens might facilitate use of a single nasal dose.

That nasal boosting with spores displaying no antigens (i.e., WT spores) evoke mucosal sIgA in primed animals is intriguing. Although the levels of induction are not as high as those in animals boosted with antigen-presenting spores this observation is important since it is possible that spores alone might provide a mechanism for evoking mucosal immunity. We are now addressing this in expanded animal and human studies.

## Figures and Tables

**Figure 1 vaccines-10-01900-f001:**
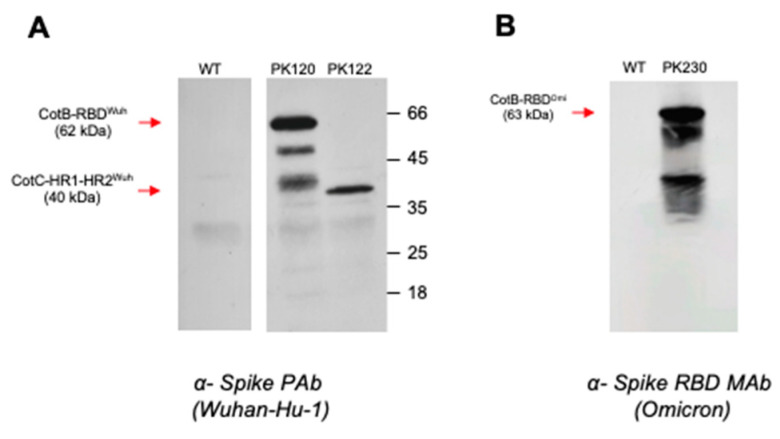
Spore coat expression of SARS-CoV-2 proteins. Panel (**A**) PK120 (*thyA::cotB-RBD^Wuh^*) and PK122 (*thyA::cotC-HR1-HR2^Wuh^*) express the RBD and HR1-HR2 domains of Spike from the Wuhan-Hu-1 (2019-nCoV) variant when probed with anti-S (Wuhan-Hu-1) PAbs. Bands corresponding to CotB-RBD^Wuh^ (62 kDa) and CotC-HR1-HR2^Wuh^ (40 kDa) are shown. No bands were detectable in spores of the isogenic strain WT which expresses no heterologous antigens. Panel (**B**) PK230 (*thyA::cotB-RBD^Omi^*) expresses the RBD domain of Omicron (B.1.1.529) fused to CotB (63 kDa) when probed with anti-S (Omicron) PAbs. No bands were detectable in spores of the strain WT.

**Figure 2 vaccines-10-01900-f002:**
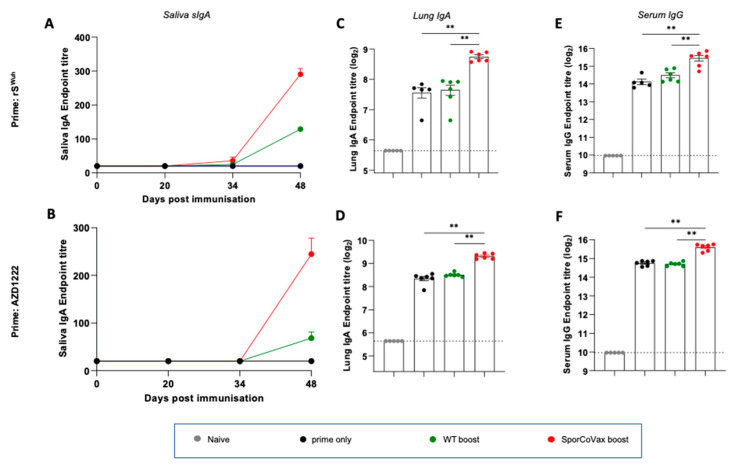
Intranasal boosting of a subunit prime or AZD1222. Female BALB/C mice were primed (i.m.) with recombinant Spike (rS^Wuh^) or AZD1222 (1.0 × 10^8^ IU) followed by two intranasal boosts with 1 × 10^9^ spores of either WT (naked spores, no antigen expression) or SporCoVax (1:1 mixture of CotB-RBD^Wuh^) and CotC-HR1-HR2^Wuh^ spores) three and five weeks post prime. Panels show S^Wuh^-specific responses determined by ELISA (OD_450 nm_) 48 days post immunization. Panel (**A**,**B**) rS^Wuh^-specific sIgA in longitudinal saliva samples, (**C**,**D**) rS^Wuh^-specific IgA in the lungs, (**E**,**F**) rS^Wuh^-specific IgG in serum. Mann–Whitney, ** *p* < 0.01.

**Figure 3 vaccines-10-01900-f003:**
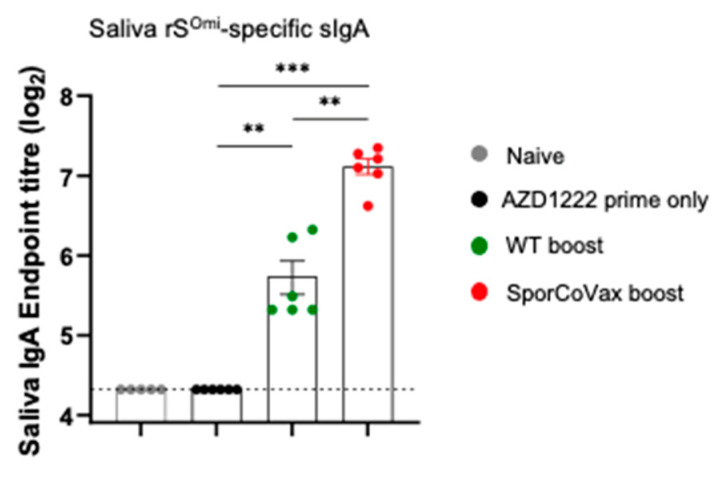
Spore-induced mucosal Spike-specific sIgA cross-reacts with Wuhan and Omicron variants. Mice were primed (i.m.) with 1.0 × 10^8^ IU of AZD1222 followed by two intranasal boosts with spores (1 × 10^9^ CFU) of WT (naked) or SporCoVax (1:1 mixture of CotB-RBD^Wuh^ and CotC-HR1-HR2^Wuh^ spores) at three and five weeks post prime. rS^Omi^-specific sIgA in the saliva 48 days post immunization is shown. Mann–Whitney, ** *p* < 0.01, *** *p* < 0.001.

**Figure 4 vaccines-10-01900-f004:**
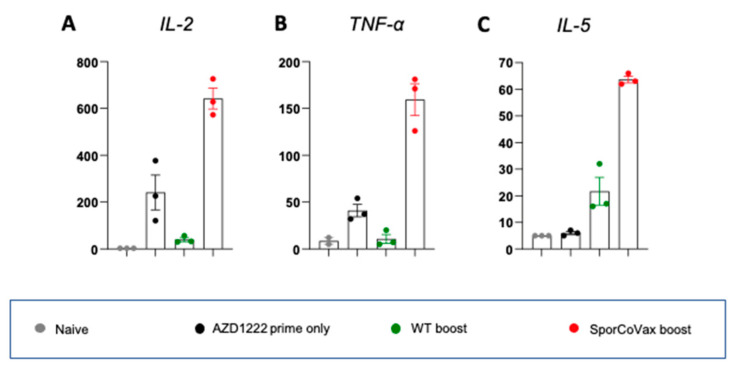
Cytokine profiles. BALB/C mice (female, aged 8 weeks; n = 3/gp) were immunized with AZD1222 (i.m.; 1.0 × 10^8^ IU) followed by two intranasal boosts with spores (1 × 10^9^ CFU/dose) of WT or SporCoVax (a 1:1 mixture of CotB-RBD^Wuh^ and CotC-HR1-HR2^Wuh^) three and five weeks post prime immunization. Control groups included untreated animals (naïve) and animals primed only (AZD1222). All mice were sacrificed 48 days post prime, and their spleens dissected and stimulated with 2.5 µg/mL rS^Wuh^ protein for 72 h, and levels of the cytokines IL-2 (panel (**A**)), TNF-α (panel (**B**)) and IL-5 (panel ((**C**)) determined by flow cytometry.

**Figure 5 vaccines-10-01900-f005:**
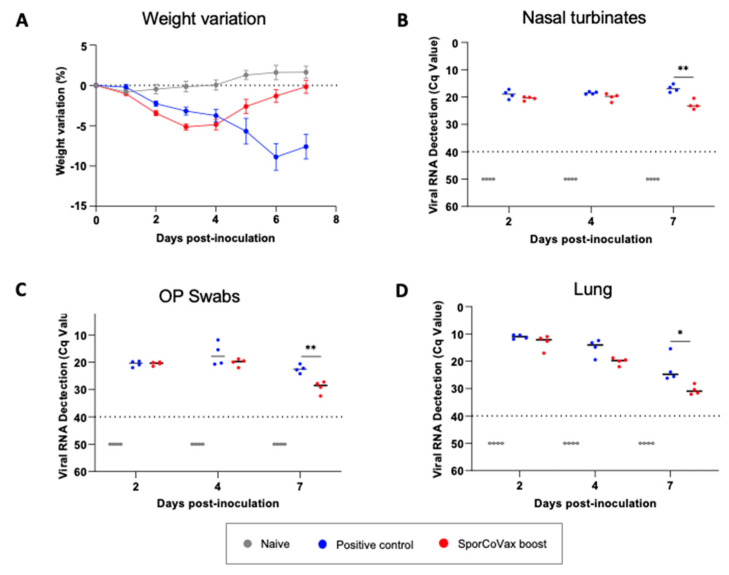
Protection in golden Syrian hamsters. Golden Syrian Hamsters were primed (i.m.) with recombinant spike (rS^Wuh^) protein followed by two intranasal boosts with 2.5 × 10^9^ spores of SporCoVax (1:1 mixture of CotB-RBD^Wuh^ and CotC-HR1-HR2^Wuh^ spores) (Group 3). Negative, unvaccinated, (Group 2) and naive (Group 1) control groups received PBS in place of immunogens. 7-days after the final boost animals in Groups 2 and 3 were intranasally challenged with SARS-CoV-2 (D614G variant, 10^4^ TCID_50_/animal). Weight loss post-challenge are shown (panel (**A**)). Animals were sequentially euthanized (2, 4, and 7 dpi), necropsies were performed, and viral load of SARS-CoV-2 were determined by qPCR in NT (panel (**B**0), OP swabs (panel (**C**)) and lungs (panel (**D**0). Mann-Whitney, * *p* < 0.05, ** *p* < 0.01.

## Data Availability

All data generated or analyzed during this study are included in this published article (and its Appendix A).

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
