# Peer review of "Heterologous Systemic Prime–Intranasal Boosting Using a Spore SARS-CoV-2 Vaccine Confers Mucosal Immunity and Cross-Reactive Antibodies in Mice as well as Protection in Hamsters"

_vaccines, 2022, doi:10.3390/vaccines10111900_

Round 1

Reviewer 1 Report

The presented manuscript represents a very interesting and complex piece of work on the preparation of engineered B. subtilis spores to express SARS-CoV-2 antigens that were used to nasally boost a systemic prime consisting of either a recombinant Spike protein or the Oxford AZD1222 vaccine. Surely, it would be good to have an additional characterization of recombinant spores as verifying that RBD and HR1-HR2 domains are displayed on the spore surface by immunofluorescence microscopy. However and importantly this work shows that the heterologous systemic prime–mucosal spore boost strategy evokes mucosal immunity and is protective in hamsters. The experimental work presents a very well-executed analysis. The reached results have wider implications for additional strategies how to prepare new vaccines against SARS-CoV-2. The discussion is detailed and very reasonable.

Author Response

Thank you very much for this encouraging review.

Reviewer 2 Report

Dear colleagues,

The title of your manuscript should be changed to include the results of your experiments in mice. Similarly, the abstract does not mention any experimental results performed in mice. In Fig. 5, I think the results for the naive and the positive control hamsters could be more clearly expressed by changing the color of the various points of one of these two groups to something like blue or green; as the dark and lighter points, both black, are not terribly clear (at least in my copy).  In general, congrats on your high-quality paper and on the exciting and promising results generated both in mice and hamsters. I think you may have introduced a very interesting approach to providing better protective immunity against SARS-CoV-2 infection and have opened the way for other scientists to expand your findings in this field.      

Author Response

Thank you for the review and we accept your recommendation and have changed the title to "Heterologous Systemic Prime - Intranasal Boosting using a Spore SARS-CoV-2 Vaccine Confers Mucosal Immunity and Cross-Reactive Antibodies in mice as well as Protection in Hamsters

The abstract has been changed to mention mice, thank you

We have also reconfigured the figures and their colour symbols to make them clearer as per your suggestion.

Reviewer 3 Report

In this work, the authors propose bacterial endospores of Bacillus subtilis engineered to display heterologous antigens as an intranasal (IN) Covid 19 boost. A heterologous systemic prime - mucosal boost regimen is used, and the objective of the work is to determine whether IN administration of spore vaccines expressing SARS-Cov-2 antigens evoke mucosal immunity and provides protection to SARS-CoV-2 infection. BALB/c mice are first vaccinated via IM with a recombinant Spike (rSWuh) or the AZD1222 (ChAdOx1) vaccine and then boosted IN with the spores. Results show that the boosted group secreted IgA in saliva and lungs against D614G variant and in saliva against omicron variant (showing cross reactivity), and cultured splenocytes produced cytokines IL-2, TNF-a and IL-5.

Please discuss whether only one IN boost is not enough to induce a response in mice.

Clarify why is necessary to apply three IN administrations for each boost in mice (and how this would be addressed in humans if needed).

In the hamster model of infection a similar vaccination regime was conducted and, after infection, weight loss was used to track progression of disease. Viral load of SARS-CoV-2 were determined by qPCR in nasal turbinate, OP swabs and lungs. Please add this in Materials and Methods section.

The major concern about this work is that the results of the challenge experiment do not show if IN vaccination protects against virus infection. The protection is indeed suggested by the lack of weight loss and the decrease in the viral load, which is not very pronounced. Authors could explain what the fate of the non-vaccinated infected animals is and how this compares to the vaccinated group. Were the animals examined for lung pathology? Do vaccinated animals finally clear the viral load in the lungs? What happens with the non-vaccinated animals that lose weight? Do they recover after day 7 or continue losing weight? is onward transmission from vaccinated animals reduced compared to that observed in unvaccinated controls? If none of these questions can be addressed, I do not find evidence to conclude that there is protection. The most accurate conclusion for the results obtained in the challenge experiment is in Line 359 of the Discussion section: “Moreover, when this strategy was employed in hamsters, they showed signs of protection from viral challenge including recovery of body weight and a reduction in viral shedding. This is encouraging for a number of reasons but most importantly a reduction in viral shedding would lead to reduced transmission and dissemination ultimately providing a means to controlling this disease.

Authors comment that “Early clinical data has suggested that without pre-existing immunity to SARS-CoV-2, immunity resulting from intranasal vaccination is suboptimal.” Hence, they propose their approach of boosting a current COVID-19 vaccine with an intranasal vaccine. I agree that this supports the lack of a group only vaccinated intranasally in the experiments.

Minor corrections

Line 141 (Boost 1, days 21, 22, 27 and Boost 2, days 35, 36, 37). (23)

Line 260 pression) ot SporCoVax (OR)

Figure 5 naive and positive control colors are too similar

Line 372 is not fully understand (understood)

Revise sentence Line 380-382: Although the mucosal responses are clearly greater when spores-displaying antigens are de- livered the use of non-recombinant spores may, long-term, have greater utility.

Author Response

Thank you for your review. I have extracted the relevant questions/comments you raise (in BOLD) and address below in italics

Please discuss whether only one IN boost is not enough to induce a response in mice.

We have added this sentence to cover this useful comment

Based on the observed seroconversion of sIgA in mice we suspect that a single dose of spore vaccine using this regimen (3-weeks post-prime) is unlikely to be protective but further empirical analysis of dosing regimens might facilitate use of a single nasal dose.

Clarify why is necessary to apply three IN administrations for each boost in mice (and how this would be addressed in humans if needed).

I have added a sentence that explains this in results.  For murine studies a single dose consisted of three daily intranasal administrations (e.g., days 21, 22 and 23) and resulted from the regulatory dosing limitations required for mice.

In the hamster model of infection a similar vaccination regime was conducted and, after infection, weight loss was used to track progression of disease. Viral load of SARS-CoV-2 were determined by qPCR in nasal turbinate, OP swabs and lungs. Please add this in Materials and Methods section.

This has been added to methods.

The major concern about this work is that the results of the challenge experiment do not show if IN vaccination protects against virus infection. The protection is indeed suggested by the lack of weight loss and the decrease in the viral load, which is not very pronounced. Authors could explain what the fate of the non-vaccinated infected animals is and how this compares to the vaccinated group. Were the animals examined for lung pathology? Do vaccinated animals finally clear the viral load in the lungs? What happens with the non-vaccinated animals that lose weight? Do they recover after day 7 or continue losing weight? is onward transmission from vaccinated animals reduced compared to that observed in unvaccinated controls? If none of these questions can be addressed, I do not find evidence to conclude that there is protection.

Vaccines are expected to protect against clinical disease, and it is clear based on weight evolution that the vaccines does the job; what the vaccine is not providing is sterilizing immunity, and in fact, it was not expected to do so. We know that weight loss is related with pneumonia severity (see Brustolin et al., 2021 for example, but many other papers), so, although pathology was not performed here, it is logical to think that the inflammatory lung lesion was the responsible of weight loss.

 On the other hand, the reviewer is asking what happened beyond day 7 post inoculation, and this is simply not possible to foresee, since the experiment ended up that day. But based on genomic RNA it is clear that the viral amount in lung was lower in vaccinated ones, which fits well again with lower weight loss and, probably, less lesions. However, we have amended the last sentence of results as follows

Taken together, we conclude that the heterologous systemic prime-intranasal boost regimen ameliorates the clinical and virological outcome of SARS-CoV-2 challenge.

Minor corrections

Line 141 (Boost 1, days 21, 22, 27 and Boost 2, days 35, 36, 37). (23) CORRECTED

Line 260 pression) ot SporCoVax (OR) WHAT WE HAVE WRITTEN IS CORRECT, IT MUST READ NO

Figure 5 naive and positive control colors are too similar, WE HAVE CHANGED

Line 372 is not fully understand (understood) CORRECTED

Revise sentence Line 380-382: Although the mucosal responses are clearly greater when spores-displaying antigens are de- livered the use of non-recombinant spores may, long-term, have greater utility. CORRECTED